Research

 

**Subject Area:**
biochemistry

Mcm10, fluorescence polarization, suramin, RPA70, surface plasmon resonance

**Authors for correspondence:**
Jon E. Hawkinson
e-mail: hawkinso@umn.edu
Anja-Katrin Bielinsky
e-mail: bieli003@umn.edu

# The anti-parasitic agent suramin and several of its analogues are inhibitors of the DNA binding protein Mcm10

Carolyn N. Paulson[1], Kristen John[1], Ryan M. Baxley[2], Fredy Kurniawan[2], Kayo Orellana[2], Rawle Francis[1], Alexandra Sobeck[2], Brandt F. Eichman[3], Walter J. Chazin[4], Hideki Aihara[2], Gunda I. Georg[1], Jon E. Hawkinson[1] and Anja-Katrin Bielinsky[2]

[1]Department of Medicinal Chemistry and Institute for Therapeutics Discovery & Development, College of Pharmacy, University of Minnesota, Minneapolis, MN 55414, USA
[2]Department of Biochemistry, Molecular Biology and Biophysics, College of Biological Sciences, University of Minnesota, Minneapolis, MN 55455, USA
[3]Departments of Biological Sciences and Biochemistry, Center for Structural Biology, Vanderbilt University, Nashville, TN 37232, USA
[4]Departments of Biochemistry and Chemistry, Center for Structural Biology, Vanderbilt University, Nashville, TN 37240, USA

A-KB, 0000-0003-1783-619X

Minichromosome maintenance protein 10 (Mcm10) is essential for DNA unwinding by the replisome during S phase. It is emerging as a promising anti-cancer target as *MCM10* expression correlates with tumour progression and poor clinical outcomes. Here we used a competition-based fluorescence polarization (FP) high-throughput screening (HTS) strategy to identify compounds that inhibit Mcm10 from binding to DNA. Of the five active compounds identified, only the anti-parasitic agent suramin exhibited a dose-dependent decrease in replication products in an *in vitro* replication assay. Structure–activity relationship evaluation identified several suramin analogues that inhibited ssDNA binding by the human Mcm10 internal domain and full-length *Xenopus* Mcm10, including analogues that are selective for Mcm10 over human RPA. Binding of suramin analogues to Mcm10 was confirmed by surface plasmon resonance (SPR). SPR and FP affinity determinations were highly correlated, with a similar rank between affinity and potency for killing colon cancer cells. Suramin analogue NF157 had the highest human Mcm10 binding affinity (FP $K_i$ 170 nM, SPR $K_D$ 460 nM) and cell activity (IC$_{50}$ 38 μM). Suramin and its analogues are the first identified inhibitors of Mcm10 and probably block DNA binding by mimicking the DNA sugar phosphate backbone due to their extended, polysulfated anionic structures.

# 1. Introduction

Minichromosome maintenance protein 10 (Mcm10) is an essential replication factor first identified in budding yeast over 30 years ago [1]. The core of Mcm10 harbours the evolutionarily conserved and essential internal domain (ID), which is composed of an oligonucleotide/-saccharide (OB-fold) and an adjacent zinc finger (ZnF) domain [2–6]. The ID is connected to the N-terminal (NTD) and C-terminal (CTD) domains by flexible linkers, highlighting the modular structure of Mcm10 [7]. The CTD is metazoan-specific and contains a second ZnF motif that facilitates DNA binding [1,8–10]. The overall absence of any known catalytic domains is consistent with the notion that Mcm10 acts as a DNA binding scaffold [1,8,9].

royalsocietypublishing.org/journal/rsob    Open Biol. 9: 190117

Mcm10 associates with DNA regardless of sequence context and topology [6,11]. It binds both double-stranded (ds) and single-stranded (ss) DNA with similar affinity. The NTD facilitates protein oligomerization, but does not exhibit any DNA binding activity [5]. Mcm10-ID displays high sequence similarity among species, and is 81% identical between *Xenopus laevis* (x) and human (h) [6]. Crystallographic studies of the xMcm10-ID have revealed that the OB-fold and ZnF domains are configured in a unique orientation that is not found in any other DNA binding protein [6]. In Mcm10, both motifs form a continuous DNA binding surface [6]. The residues identified by nuclear magnetic resonance chemical shift perturbation to make contact with DNA span a patch of basic lysines and aromatic amino acids [6]. Although the CTD ZnF significantly increases the DNA binding affinity of full-length Mcm10, small molecules that bind ZnF domains are unlikely to lead to selective drugs due to the similarity of ZnF domains in diverse proteins [12–14], suggesting that the ID alone is the best target for Mcm10-specific inhibitors.

There is accumulating evidence that Mcm10 plays an important role in cancer development. Many studies have reported *MCM10* overexpression in a variety of cancer types [1,15–18]. Furthermore, the level of *MCM10* upregulation in some cancers has been correlated with tumour progression or poor clinical outcomes [1,15,19]. Consistent with expression studies, cancer genome analyses reveal that the majority of chromosomal changes are gene amplifications, whereas *MCM10* is rarely deleted [1]. Given that *MCM10* has been identified as a key suppressor of DNA damage in several studies, including two independent genome-wide screens, it has been hypothesized that cancer cells rely on high levels of Mcm10 to promote growth and reduce genome instability [1,20–22].

Taken together, Mcm10 appears to be a promising anti-cancer drug target. To date, chemical inhibitors of Mcm10 have not been reported and only a handful of drugs have been isolated that interfere with protein-DNA binding [23,24]. Because Mcm10 function is tied to DNA binding, a high throughput screen was performed to elucidate inhibitors of Mcm10 that interfere with its binding to ssDNA. The anti-microbial agent suramin and several of its analogues were found to inhibit the DNA binding activity of human Mcm10 internal domain (hMcm10-ID) with affinity values ranging from 0.17 to 77 µM by fluorescence polarization (FP) and similar values by surface plasmon resonance (SPR).

# 2. Material and methods

## 2.1. Mcm10 fluorescence polarization HTS assay

A fluorescence polarization (FP) HTS assay was established to detect inhibitors of the binding of the 5′-6FAM 10-mer oligo probe to Mcm10. Test compounds dissolved in DMSO were added to 384-well plates (Corning 4514) using an Echo 550 acoustic dispenser (final DMSO 0.1%) to achieve a final single point screening concentration of 10 µM. Then 10 µl of 2× *Xenopus* internal domain Mcm10 (xMcm10-ID, final concentration 2 µM) in binding buffer (20 mM Tris HCl, 100 mM NaCl, 5% glycerol, and 0.01% triton, pH 7.5)

was added using the Combi nL Multidrop dispenser. Finally, 10 µl of 2× 5′-6FAM DNA (final 12.5 nM) in binding buffer was added using the Multidrop. DMSO (0.1%, high signal) and probe only (low signal) controls were included on every plate. Plates were mixed for 2 min, incubated for 60 min at RT in the dark, and read on a CLARIOstar multi-mode plate reader (excitation: 482-16, emission: 530-40, dichroic filter: LP504).

## 2.2. *Xenopus* egg extract preparation and *in vitro* DNA replication assay

*Xenopus* egg extracts were prepared according to the method of Murray [25,26]. Replication of sperm chromatin in S-phase egg extracts was monitored as previously described [25]. Compounds identified by HTS were added to replication reactions immediately prior to addition of [α-$^{32}$P]dGTP (Perkin Elmer BLU514H250UC).

## 2.3. Surface plasmon resonance

SPR experiments were performed using a Biacore S200 (GE Healthcare) equipped with a research-grade CM5 sensor chip. hMcm10-ID, at a concentration of 15 µg ml$^{-1}$ in 10 mM sodium acetate, pH 4.5, was immobilized at a density of 6000–8000 RU, using an amine coupling kit (GE Healthcare) to either flow cell 2 or 4 following manufacturer directions. The reference flow cell (either flow cell 1 or 3) was left untreated. All compounds were dissolved in running buffer (10 mM PBS, 150 mM NaCl, 0.005% P20, pH 7.4) prior to injection over the chip surface at a flow rate of 30 µl min$^{-1}$ and at a temperature of 25°C. Zero concentration samples were injected for double referencing. Data were collected at a rate of 40 Hz and were fitted to a simple 1 : 1 interaction model (unless otherwise noted) using the global data analysis within the Biacore S200 evaluation software.

## 2.4. Cell viability assay

HCT116 cells were grown in McCoy's 5A medium (Corning 10-050-CV) supplemented with 10% FBS (Sigma F4135), 1% Pen Strep (Gibco 15140) and 1% L-Glutamine (Gibco 205030). hTERT RPE-1 cells were grown in DMEM/F12 medium (Gibco 11320) supplemented with 10% FBS (Sigma F4135) and 1% Pen Strep (Gibco 15140). Cells were cultured at 37°C and 5% CO$_2$. Cells were plated at 250 cells (hTERT RPE-1) or 500 cells (HCT116) per well in white-walled 96-well plates (Costar 3610) and allowed to recover for 24 h. Stock solutions of each inhibitor were prepared in sterile 1× PBS (Gibco 14190), and further diluted in the appropriate growth medium for each cell type. Cells were allowed to grow for 4 days in inhibitor containing medium and cell viability was measured with the CellTiter-Glo Luminescent Cell Viability Assay (Promega G7572) following the manufacturer's instructions. Viability of each drug treatment condition was normalized to the untreated control for each cell line and fitted to the sigmoidal dose–response variable slope four parameter equation in GraphPad Prism 6.0.

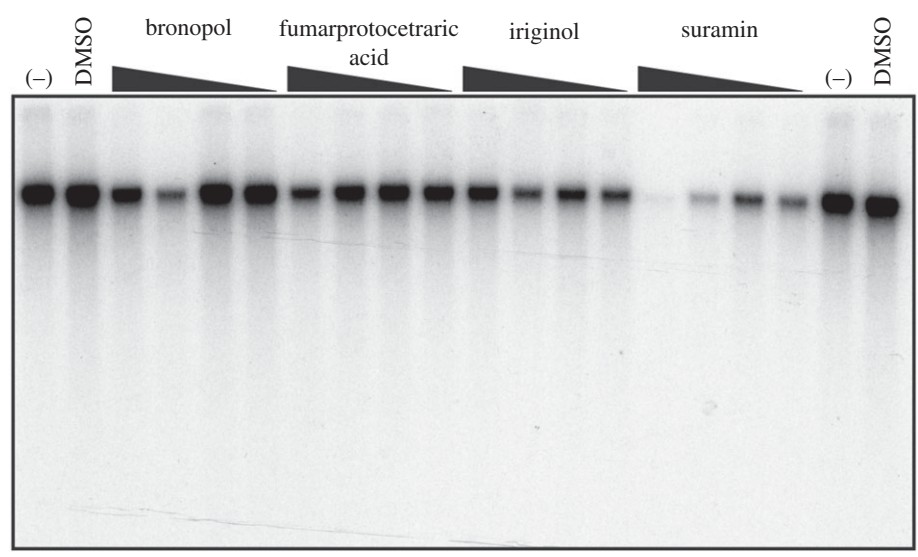

**Figure 1.** Suramin inhibits DNA replication in an *in vitro Xenopus* replication assay. HTS hits were evaluated for inhibition of [$\alpha$-$^{32}$P]dGTP incorporation into DNA using a cell-free extract prepared from *Xenopus* eggs at increasing concentrations (10, 50, 150 and 300 µM indicated by black triangle from right to left). No chemical (−) and DMSO were used as negative controls for replication inhibition (far left and far right lanes).

# 3. Results

## 3.1. High throughput screening identifies five active inhibitors

A previously described FP assay for determining binding affinities of ssDNA to Mcm10 [5] was used to screen compound libraries for inhibitors of the binding of a 5′-6-carboxyfluorescein 10-mer DNA (5′-6-FAM-ATGGTAGGCA) (5′-6FAM) probe to xMcm10-ID. The 5′-6FAM DNA probe binds xMcm10-ID with a $K_D$ of 1.3 µM (electronic supplementary material, figure S1 and table S1). A pilot screen of the Library of Pharmacologically Active Compounds (LOPAC) using a 3′- rather than 5′-6FAM provided an average $Z'$ value of 0.61. Although $Z'$ scores greater than or equal to 0.5 are indicative of a robust assay suitable for high-throughput screening (HTS), the 5′-6FAM probe was subsequently evaluated in an effort to improve assay robustness and shown to provide a higher signal window with a $Z'$ of 0.90. Although speculative, the higher $\Delta$mP value provided by 5′ attachment of the fluorophore to the oligo may be due to better engagement of the fluorophore with Mcm10 in this position, resulting in reduced 'propeller effect' [27]. An HTS of greater than 150 000 compounds was conducted with the 5′-6FAM probe (mean $Z'$ value for entire screen was 0.89) (electronic supplementary material, figure S2). A low threshold of 25% inhibition was selected due to the low hit rate. This low hit rate is not surprising as DNA binding proteins, such as transcription factors, are generally considered to be undruggable [28]. A total of 158 compounds produced greater than or equal to 25% inhibition of the mP signal, providing a primary hit rate of 0.10%. However, only 39 of these compounds exhibited little effect on total fluorescence intensity, suggesting lack of fluorescence interference, resulting in a low effective hit rate of 0.025%. Based on dose–response studies of these 39 hits cherry-picked from DMSO stocks, 11 were weak or inactive (IC$_{50}$ values > 100 µM), five exhibited structural alerts (e.g. reactive, polymers) and 16 had substantial fluorescence interference. The seven remaining hits were repurchased: two were found to be inactive or produce partial inhibition and five

were active. The five confirmed HTS hits were: suramin sodium salt, a poly-sulfated napthylamine originally developed in the 1920s to treat African trypanosomiasis [29]; bronopol, a brominated di(hydroxymethyl) nitromethane that is used as an anti-bacterial agent and pharmaceutical preservative [30]; fumarprotocetraric acid, a lichen derived depsidone reported to have anti-microbial, anti-carcinogenic, antioxidant, and immunostimulatory properties [31]; the natural product derivative iriginol hexaacetate, which has been shown to inhibit a bacterial ribonuclease [32]; and 4-chloromercuribenzoic acid (PCMB), a cysteine active site modifier that inhibits some enzymes requiring unmodified cysteine residues for activity (e.g. adenylyl cyclases) [33]. Although initially used as a positive control during the HTS, PCMB was eliminated from further studies because it probably covalently reacts with surface cysteine residues to block DNA binding to Mcm10.

## 3.2. Suramin inhibits DNA replication *in vitro*

To determine whether the four remaining confirmed HTS hits affected DNA replication, *in vitro* replication assays were performed using *Xenopus* S-phase egg extracts incubated with sperm chromatin [25,34]. DNA replication was measured in the presence of bronopol, fumarprotocetraric acid, iriginol and suramin or DMSO by the incorporation of [$\alpha$-$^{32}$P]dGTP. Of the HTS hits tested, only suramin exhibited a dose-dependent decrease in DNA replication (figure 1). For this reason, further studies focused on suramin alone.

## 3.3. SAR and selectivity of Mcm10 inhibitors

About 30 commercially available suramin analogues and smaller sulfated polycyclic organic compounds were identified and purchased to establish the structure–activity relationships (SAR) for suramin-like compounds at hMcm10-ID using the FP assay. Similar to xMcm10-ID, the 5′-6FAM DNA oligo probe bound hMcm10-ID with low micromolar affinity (electronic supplementary material, figure S1 and table S1). While the lower molecular weight

**Figure 2.** Structures of suramin and suramin analogues.

analogues were weakly active or inactive ($K_i > 100\,\mu M$), nine of the poly-sulfated, high-molecular-weight compounds displaced the FP probe with $K_i$ values < 100 $\mu M$ (structures are shown in figure 2). The active suramin analogues displayed a broad range of potencies for hMcm10-ID with $K_i$ values ranging from 170 nM to 77 $\mu M$ (table 1; electronic supplementary material, figure S3A).

Having confirmed that suramin and analogues bind with a range of affinities to the hMcm10-ID, this series of compounds was then tested with full-length *Xenopus* Mcm10 (xMcm10-FL) to determine if they also bind to the full-length protein. xMcm10-FL was chosen for these studies instead of hMcm10-FL because the human protein is significantly less stable. The affinity of the 5′-6FAM DNA oligo for the xMcm10-FL protein was 1.1 $\mu M$ (electronic supplementary material, figure S1 and table S1) prior to testing compounds. The larger compounds were active towards both proteins but displayed higher affinity for hMcm10-ID over xMcm10-FL, up to 20-fold in the case of NF157. Interestingly, *iso*-pyridoxalphosphate-6-azophenyl-2′, 4′-disulfonic acid (*iso*-PPADS) and PPADS inhibited probe binding to xMcm10-FL by less than 50% when added up to a concentration of 2 mM (table 1; electronic supplementary material, figure S3B). The higher potencies of the compounds for hMcm10-ID over xMcm10-FL could be due to overall conformational changes between the full-length protein and the internal domain or, alternatively, species differences between the human and *Xenopus* proteins. To explore this, *iso*-PPADS and PPADS were tested with xMcm10-ID and found to possess little activity toward xMcm10-ID (electronic supplementary material, figure S5), suggesting that these compounds preferentially bind human over *Xenopus* Mcm10.

To investigate the selectivity of suramin and its analogues for Mcm10 relative to other DNA binding proteins involved in DNA replication, the affinity of these compounds for human replication protein A (RPA) was determined. We expressed a construct containing both the A and B DNA binding domains of RPA70 (RPA70AB; table 1; electronic supplementary material, figure S3C). RPA70AB is the tandem high-affinity ssDNA binding domain, which plays an important role in DNA replication and repair [35]. The 5′-6FAM DNA probe had a $K_D$ of 0.4 $\mu M$ for RPA70AB (electronic supplementary material, figure S1 and table S1). The compound with the highest affinity for hMcm10-ID, NF157, was 11-fold selective for hMcm10 over RPA70AB. Complicating this selectivity analysis was the finding that certain compounds (NF449, NF110, and NF546) produced biphasic dose–response curves for displacement of the probe from RPA70AB. Although the mechanism of this two component displacement is unknown, one explanation is that these compounds displace the probe from the A and B domains with differential affinity. However, this explanation assumes that the probe binds the A and B domains independently and with similar affinities, which does not fit with the proposed sequential DNA binding model in which the A and B domains bind simultaneously to the DNA strand due to the short linker between them [35]. Considering the predominant low-affinity component, NF449 was the most selective of these biphasic compounds for Mcm10 over RPA (32-fold). Despite being weaker inhibitors of Mcm10, PPADS and *iso*-PPADS were the most selective (greater than 50-fold) for hMcm10 as they were very weak inhibitors of RPA70AB.

## 3.4. Validation of binding kinetics by surface plasmon resonance

To confirm the binding of suramin and analogues to Mcm10 using an orthogonal method, $K_D$ values were obtained using

**Table 1.** Affinity of suramin analogues for Mcm10 determined by SPR and FP: selectivity for Mcm10 over RPA.[a]

| compound | fluorescence polarization | | | | selectivity | surface plasmon resonance | | |
| | hMcm10-ID | xMcm10-FL | hRPA70AB | | | hMcm10-ID | | |
| | $K_i$, μM | $K_i$, μM[b] | $K_{1r}$, μM | $K_{2r}$, μM | RPA/hMcm10 | $k_a$ ($M^{-1}s^{-1}$)[c] | $k_d$ ($s^{-1}$)[c] | $K_D$ (μM)[c] |
| --- | --- | --- | --- | --- | --- | --- | --- | --- |
| NF157 | 0.17 ± 0.02 | 3.4 | — | 1.9 ± 0.1 | 11 | $7.4 ± 0.6 × 10^3$ | 0.0032 ± 0.0001 | 0.46 ± 0.05 |
| NF279 | 0.33 ± 0.05 | 2.3 | — | 0.63 ± 0.03 | 2 | $8.5 ± 0.2 × 10^5$ | 0.059 ± 0.007 | 0.74 ± 0.10 |
| NF449 | 0.44 ± 0.04 | 1.9 | d | 14.0 ± 0.4 | 32 | $7.8 ± 0.1 × 10^5$ | 0.59 ± 0.10 | 0.74 ± 0.13 |
| suramin | 0.83 ± 0.07 | 3.4 | — | 2.5 ± 0.1 | 3 | $2.9 ± 0.7 × 10^5$ | 0.16 ± 0.03 | 0.57 ± 0.06 |
| NF023 | 1.8 ± 0.2 | 4.4 | — | 11 ± 1 | 6 | $3.6 ± 0.2 × 10^5$ | 1.6 ± 0.2 | 4.7 ± 0.8 |
| NF110 | 2.3 ± 0.1 | 2.9 | 0.21 ± 0.04 | 23 ± 1 | 10 | $2.9 ± 1.6 × 10^5$ | 0.35 ± 0.19 | 1.5 ± 0.2 |
| NF546 | 8.1 ± 0.5 | ND[e] | 0.66 ± 0.14 | 130 ± 9 | 16 | two component binding[f] | | |
| PPADS | 17 ± 1 | >1000 | — | >1000 | >59 | 29 ± 8 | 0.0013 ± 0.0001 | 57 ± 9 |
| iso-PPADS | 18 ± 2 | >1000 | — | >1000 | >56 | 47 ± 8 | 0.0015 ± 0.0001 | 40 ± 9 |
| NF340 | 77 ± 7 | ND[g] | ND[g] | ND[g] | ND[g] | ND[g] | ND[g] | ND[g] |

[a]Mean ± s.e.m., $n ≥ 3$ unless otherwise noted.

[b]Values are averages of $n = 2$ due to limited protein availability.

[c]Binding affinity of suramin analogues for hMcm10-ID was measured by surface plasmon resonance (SPR) and kinetic $K_D$ values were calculated from association and dissociation rate constants using the Biacore S200 Evaluation Software. The kinetic $K_D$ value for suramin of 0.65 μM obtained at a lower surface density of 3000 RU was similar to the value obtained at 6000 RU (electronic supplementary material, figure S4), indicating that the higher surface density used in these studies did not affect the measured affinity values.

[d]The $IC_{50}$ value for the high affinity component (0.093 ± 0.005 μM) was below lower limit of the sensitivity of the assay.

[e]Not determined due to limited protein availability.

[f]See Results for kinetic parameters and electronic supplementary material, figure S8.

[g]Not determined due to lack of available compound.

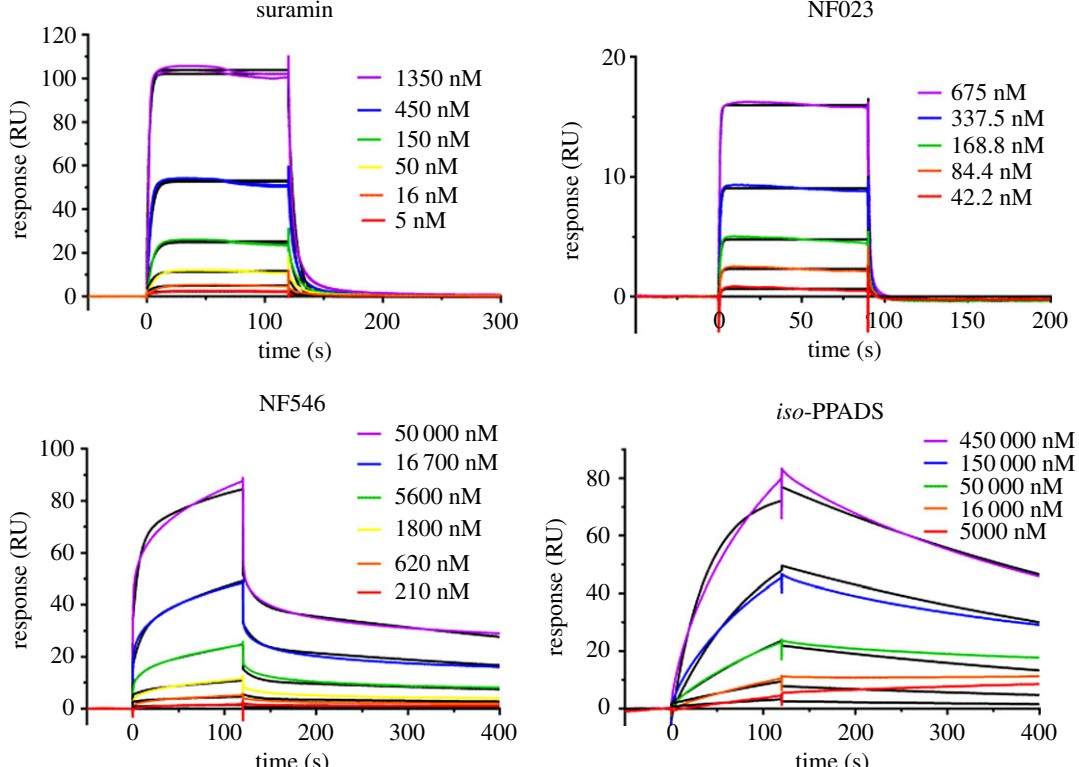

**Figure 3.** Binding kinetics of suramin, NF023, NF546, and *iso*-PPADS to immobilized hMcm10-ID. Note the rapid association and dissociation of NF023 in contrast to the slow on- and off-rates of *iso*-PPADS. NF546 displays slow and rapid components in both the association and dissociation phases, whereas suramin has predominantly rapid kinetics. Each figure is a representative SPR sensorgram from one of more than three experiments.

SPR (figure 3; table 1; electronic supplementary material, figure S6). All sensorgrams were fitted to a one-to-one binding surface model, except for NF546. Overall, the resulting kinetic $K_D$ values correlated well with the $K_i$ values determined from FP (electronic supplementary material, figure S7). The most potent compounds, NF157, NF279, NF449 and suramin, had submicromolar kinetic $K_D$ values ranging from 0.46 to 0.74 µM, whereas NF110 and NF023 exhibited $K_D$ values of 1.5 and 4.7 µM. PPADS and *iso*-PPADS were the least potent of the compounds tested by SPR with double digit micromolar $K_D$ values (57 and 40 µM, respectively). The NF546 kinetic data fitted poorly to the 1 : 1 Langmuir model, having larger residuals (difference between the experimental data and fitted curves) and a significantly higher $\chi^2$ value compared to the more complex models (electronic supplementary material, figure S8). Using the two-state fit, NF546 bound hMcm10-ID with $k_{a,1} = 1.38 \pm 0.08 \times 10^4 \, M^{-1} \, s^{-1}$, $k_{d,1} = 0.63 \pm 0.03 \, s^{-1}$, $k_{a,2} = 8.1 \pm 0.8 \times 10^{-3} \, M^{-1} \, s^{-1}$, $k_{d,2} = 0.0014 \pm 0.0002$, and overall kinetic $K_D = 7.0 \pm 1.6$ µM.

Inspection of the sensorgrams (figure 3; electronic supplementary material, figure S6) indicated that most of the compounds had rapid binding kinetics best exemplified by NF023. In contrast, the small, phosphate-containing binders, PPADS and *iso*-PPADS, exhibited slow on and off rates. These observations suggested the possibility that the fast and slow kinetic components might have corresponded to compound binding to two discrete sites, one with rapid and one with slow binding kinetics. To explore this possibility, competition-based SPR experiments were conducted using the ABA injection method to determine if the slow binder *iso*-PPADS competes with the fast kinetic compound NF023 (figure 4). ABA injection allows solution A (NF023 at 10× its $K_D$) to be injected over the surface with solution

B (NF023 + competitor at 1× $K_D$) in the same cycle. Competitive binding should result in little or no change in binding response between the A and B injections, whereas non-competitive binders are expected to produce an additive binding response. When NF023 at 1× $K_D$ was the competitor, a significant response was observed in the B phase (figure 4b, black line), but there was no increase in response above that produced by a 10-fold higher concentration of its $K_D$ (figure 4b, green line), consistent with a competitive interaction. Both suramin (figure 4c) and *iso*-PPADS (figure 4d) exhibited similar responses when run as competitors in that neither compound showed additive binding. The lack of additive binding during the B-phase of the SPR experiment indicated that the fast and slow kinetic inhibitors compete for the same binding site on Mcm10.

## 3.5. Suramin and its analogues preferentially kill transformed cells that overexpress Mcm10

To investigate the potential of Mcm10 inhibitors to preferentially kill cancer cells, the effect of suramin, NF157 (the most potent suramin analogue), NF546 (lower affinity, but higher selectivity), *iso*-PPADS and PPADS (low affinity, but most selective) on the survival of two cell lines of epithelial origin was tested. We selected non-transformed hTERT RPE-1 and colon cancer HCT116 cells, which—unlike hTERT RPE-1—overexpress Mcm10 (table 2; electronic supplementary material, figure S9). NF157 was the most potent cytotoxic compound tested against the cancer cell line, whereas *iso*-PPADS and PPADS were the least potent, matching their relative affinities by FP and SPR and suggesting that a correlation exists between cell killing potency and inhibition of

royalsocietypublishing.org/journal/rsob   Open Biol. 9: 190117

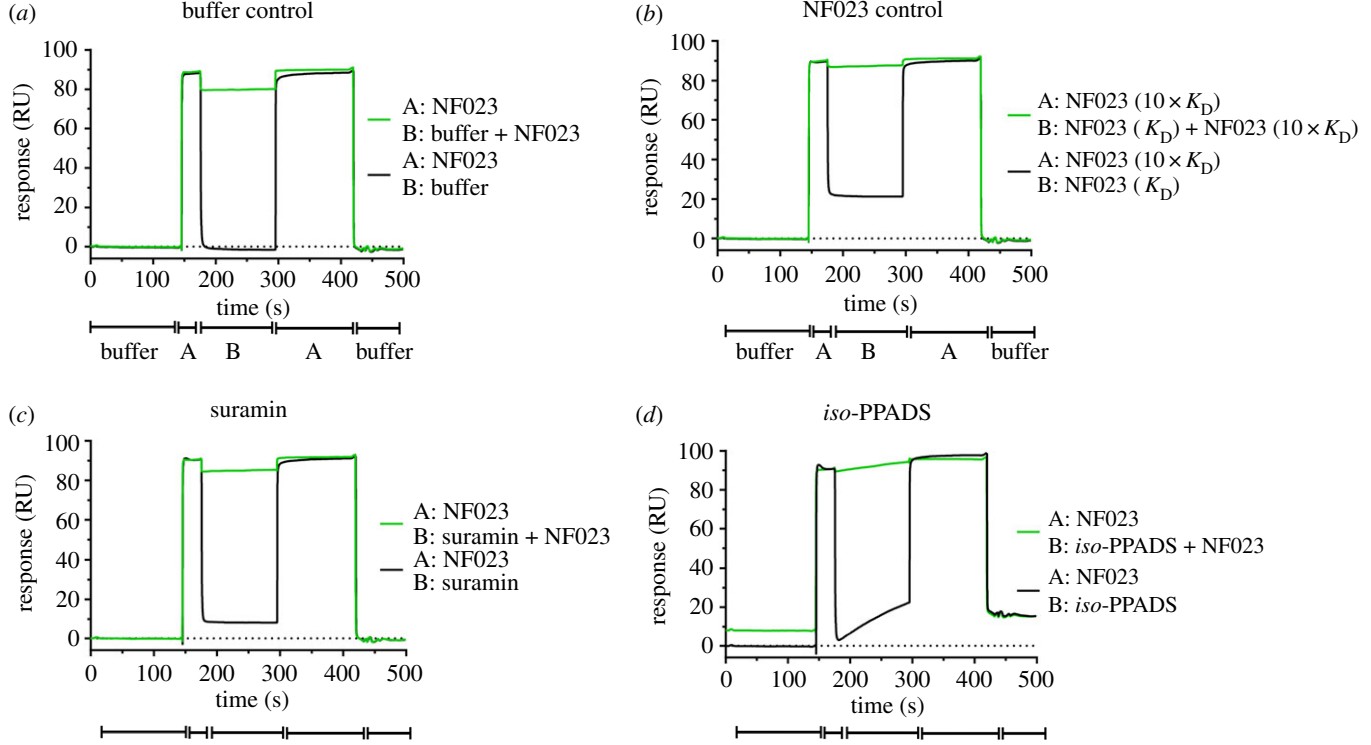

**Figure 4.** The slow binding inhibitor *iso*-PPADS competes with the fast binder NF023 for the same site on Mcm10. Representative sensorgrams of ABA SPR competition experiments ($n > 3$) in which injections of NF023 flank an injection of a 'competitor' compound: (*a*) running buffer, (*b*) NF023 self-competition, (*c*) suramin, or (*d*) *iso*-PPADS with and without NF023 at $10\times K_D$. Note that a $K_D$ concentration of NF023, suramin, and *iso*-PPADS produce the expected RU in the B phase (black line), but none produce a greater response than a $10\times K_D$ concentration of NF023 alone (green line in B phase compared to A phases), indicating that they compete with NF023 for the same site.

**Table 2.** Cytotoxicity of suramin analogues for human epithelial hTERT RPE-1 cells and colon cancer HCT116 cells.[a]

| compound | IC$_{50}$, μM[b] | | selectivity |
| --- | --- | --- | --- |
| | **HCT116** | **hTERT RPE-1** | **selectivity** |
| NF157 | 38 ± 1 | 120 ± 16 | 3.2 |
| NF546 | 91 ± 7 | 310 ± 30 | 3.4 |
| suramin | 109 ± 10 | 400 ± 22 | 3.7 |
| PPADS | 310 ± 20 | 400 ± 14 | 1.3 |
| *iso*-PPADS | 370 ± 30 | 560 ± 55 | 1.5 |

[a]Cytotoxicity of suramin analogues for hTERT RPE-1 and HCT116 cells was determined by measuring intracellular ATP concentrations. Selectivity = hTERT RPE-1 IC$_{50}$/HCT116 IC$_{50}$.
[b]Mean ± s.e.m. where $n = 3$.

Mcm10 binding to DNA. The larger, higher-affinity molecules suramin, NF157 and NF546 showed approximately 3.5-fold higher cytotoxicity for HCT116 cancer cells in comparison to hTERT RPE-1 cells, whereas the weakly cytotoxic compounds *iso*-PPADS and PPADS showed little cell type specificity.

## 4. Discussion

Suramin and its analogues have been extensively studied since the parent compound was first developed in the 1920s to treat African sleeping sickness [36]. Suramin is best known as a purinergic receptor agonist, and captured

renewed attention when reports about its ability to correct autism-like features in mice and a small phase I/II randomized clinical trial for the treatment of autism spectrum disorder were published [37–39]. In addition, suramin has shown anti-proliferative effects in several human cancer cell lines and human tumour specimens, and in animal cancer models [40–42]. Use as an anti-neoplastic agent in clinical trials has been problematic due to limited membrane permeability [40]. However, new drug delivery systems, such as the recently reported glycol chitosan-based nanoparticles, demonstrated effective treatment of lung metastases arisen from triple-negative breast cancer in mice without any cardiotoxicity or renal damage [43]. Therefore, interest in suramin and its mechanism of action remains high. Diverse protein targets for suramin have been reported, ranging from purinergic receptors to anti-viral and cancer targets [44]. Suramin, NF546 and NF157 are P2Y$_{11}$ purinergic receptor antagonists [45–47]. Recently, suramin has been shown to interfere with intracellular signalling proteins in the WNT pathway to inhibit the growth of breast cancer cells in a mouse xenograft model [48], and to target SH2 domains of STAT5a/b and STAT1 [49]. Suramin has also been reported to compete with a poly(A),oligo(dT) primer for the inhibition of reverse transcriptase [50] and to disrupt dsDNA binding to cyclic GMP-AMP synthase [51]. Despite the large number of potential targets, only a few interactions have been characterized in depth.

In the present study, we carried out a fluorescence-based DNA competition HTS using xMcm10-ID and identified seven active compounds that exhibited greater than 25% inhibition of DNA oligo binding and did not produce any significant fluorescence interference. After subsequent

royalsocietypublishing.org/journal/rsob   Open Biol. **9**: 190117

dose-dependent inhibition of chromatin replication in *Xenopus* egg extracts, suramin was the only compound of interest that not only prevented DNA binding to xMcm10-ID, but also disrupted *in vitro* DNA synthesis.

After identifying and testing several commercially available suramin analogues for hMcm10-ID affinity, we observed that the presence of phosphate groups conferred reduced activity, as three of the four least potent compounds (NF546, *iso*-PPADS and PPADS) all contain at least one phosphate or phosphonate group. Phosphate and sulfate functional groups have slightly differing Lewis acid complexation geometries that could result in differences regarding preferential hydrogen bond architecture [52], potentially explaining why the sulfate-containing compounds are more potent. The number and positioning of the sulfates on the naphthalene rings also appear to be important for activity (figure 2). NF340, the least potent compound with a $K_i$ of 77 µM, has only two sulfate groups in positions 3 and 7 of the naphthalene core. In comparison, the higher affinity compounds suramin, NF157, NF279 and NF023 all have three sulfate groups in the 4, 6 and 8 positions of the naphthalene ring. Additionally, NF449 and NF110 are identical in structure in all but the number of sulfate groups present on the benzene rings. NF449, with two sulfate groups per benzene ring, is more potent than NF110, which has only one sulfate per benzene ring. These observations, taken together with the literature studies discussed above, suggest it is likely that the long, polysulfated suramin molecule mimics the DNA sugar phosphate backbone, allowing it to bind several DNA interacting proteins including Mcm10.

As suramin and its analogues appear to mimic the DNA sugar phosphate backbone, we carried out additional dose–response activity assays with xMcm10-FL and hRPA, a related DNA binding protein, to test for selectivity. Overall, the compounds were selective for hMcm10-ID over xMcm10-FL and hRPA. Several compounds were more than 10-fold selective for hMcm10 over hRPA (NF157, NF549 and NF546) and PPADS/iso-PPDAS were greater than 50-fold selective for hMcm10, suggesting that these compounds are useful probes to selectivity inhibit the function of Mcm10 in cells. Further, dose–response studies with xMcm10-ID using iso-PPADs and PPADs suggest that the increased affinity toward hMcm10-ID is species-specific and probably not due to conformational changes between the ID and full-length proteins.

As SPR competition studies indicated that fast and slow binders interact with the same site on Mcm10, and noting the correlation between the binding kinetics and structural features of the inhibitors, we hypothesize that slow-binding small molecules containing phosphate groups and fast-binding sulfate-containing inhibitors bind to two different conformations of Mcm10. This conformation-dependent binding could be mediated through an induced-fit mechanism in which the binding event and protein conformational change occur simultaneously, or by conformational selection in which the binder interacts with only one of multiple pre-existing protein conformations. Our data is consistent with the conformational selection model in which one Mcm10 conformation is bound by fast on/fast off sulfate-containing inhibitors (e.g. NF023), and a second Mcm10 conformation is bound by slow on/slow off phosphate-containing inhibitors (e.g. *iso*-PPADS) that mimic the DNA backbone. However, binding to this second conformation may involve an induced-fit mechanism in which the slow binding kinetics

of phosphate-containing inhibitors is due to the time required for Mcm10 conformational change. In support of this theory, DNA binding induces conformational changes in Mcm10 by NMR [6]. In this context, the sulfate- and phosphonate-containing inhibitor NF546 demonstrates clear multicomponent binding kinetics (figure 3; electronic supplementary material, figure S8), suggesting that it binds both Mcm10 conformations. The two-state conformational change model is favoured over the heterogeneous ligand model as the conformation of Mcm10 is known to change on DNA binding [6] and because hMcm10-ID eluted as a single peak on gel filtration, suggesting a single population of Mcm10 was immobilized on the chip surface.

Lastly, the effects of suramin, NF157 (the most potent suramin analogue), NF546 (lower affinity, but higher selectivity), and *iso*-PPADS and PPADS (low affinity, but most selective) on colon cancer HCT116 cells and normal epithelial hTERT RPE-1 cells were evaluated (table 2; electronic supplementary material, figure S9). We found that analogue NF157 was the most potent compound tested against the cancer cell line and that overall, the high affinity molecules showed approximately 3.5-fold higher cytotoxicity for HCT116 cells over the hTERT RPE-1 cells. These results suggest that suramin and analogues NF157 and NF546 may preferentially kill cancer cells by inhibiting Mcm10 function, although activities at other proteins cannot be ruled out for these promiscuous compounds.

In summary, the anti-parasitic agent suramin and several of its analogues are potent inhibitors of the DNA binding protein Mcm10. NF157 was the most potent inhibitor and the suramin analogues displayed a range of selectivity for Mcm10 over RPA70AB. Moreover, suramin, NF157 and NF546 may preferentially kill cancer cells by blocking Mcm10-dependent DNA replication. Suramin and its analogues represent the first reported inhibitors of Mcm10 and these compounds may lead to the development of higher potency and more selective small molecules with improved physico-chemical properties targeting Mcm10 to treat cancer.

## Abbreviations

FP, fluorescence polarization; hMcm10-ID, human Mcm10 internal domain; Mcm10, mini-chromosome maintenance protein 10; RPA70AB, replication protein A 70 kD DNA-binding subunit A and B DNA binding domains; SPR, surface plasmon resonance; xMcm10-FL, full-length *Xenopus* Mcm10 protein; xMcm10-ID, *Xenopus* Mcm10 internal domain.

## Supporting information

The following can be found in the electronic supplementary material: additional FP assay details; screening collections, compounds and reagents; protein production and purification; compound purity analysis by LCMS; SPR competition and control experiments; saturation curves and affinities of the FP probe to hMcm10-ID, xMcm10-FL, and RPA70AB; FP HTS assay performance Z' plot; $IC_{50}$ curves of suramin analogues with hMcm10-ID, xMcm10-FL, and RPA70AB; inhibitory potency of PPADS and *iso*-PPADS with xMcm10-ID; SPR kinetic sensorgrams of NF 449, NF110, NF 157, NF279 and PPADS; log $K_i$ versus log $K_D$

correlation plot; comparison of binding models to NF546 SPR sensorgrams; SPR kinetic sensorgrams of suramin with a lower density surface; IC$_{50}$ curves of suramin analogues with hTERT RPE-1 and HCT116 cells.

Data accessibility. This article has no additional data.

Competing interests. The authors declare no potential conflicts of interest.

Funding. This work was supported by an Academic Health Center Faculty Research Development grant from the University of Minnesota Medical School and NIH grant no. 5R01 GM074917 to A.-K.B. This work was also supported by an NIH NIGMS grant no. R35-GM118047 to H.A., NIH ORIP grant no. 1S10OD021539 to J.E.H., and NIH grant no. R35 GM118089 to W.J.C.

Acknowledgements. We would like to thank Dr Matthew Cuellar for performing the LCMS purity analysis.

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
