## [Reviewer comments · Open Biology]

Review History

RSOB-19-0117.R0 (Original submission)

Review form: Reviewer 1

Recommendation

Accept as is

Are each of the following suitable for general readers?

- a) **Title**
Yes
- b) **Summary**
Yes
- c) **Introduction**
Yes

Is the length of the paper justified?

Yes

Should the paper be seen by a specialist statistical reviewer?

No

Is it clear how to make all supporting data available?

Yes

Is the supplementary material necessary; and if so is it adequate and clear?

Yes

Do you have any ethical concerns with this paper?

No

Comments to the Author

Suramin is an interesting compound that acts as a DNA or RNA mimic and it is turning our that suramin can be used to block several DNA protein interactions. What I like about this paper is that they do careful characterizations of he suramin/analogs interactions with the target. The anticancer activities is a bit weak but that could be because of permeation issues. In any case I recommend acceptance.

Review form: Reviewer 2

Recommendation

Accept with minor revision (please list in comments)

Are each of the following suitable for general readers?

- a) **Title**
Yes
- b) **Summary**
Yes
- c) **Introduction**
Yes

Is the length of the paper justified?

Yes

Should the paper be seen by a specialist statistical reviewer?

No

Is it clear how to make all supporting data available?

Yes

Is the supplementary material necessary; and if so is it adequate and clear?

Yes

Do you have any ethical concerns with this paper?

No

Comments to the Author

This is an interesting paper. DNA-binding proteins are considered "difficult" targets for drug discovery, so the success reported here in the identification of hits against mcm10 (albeit with a low hit rate) will be of wide interest to readers interested in such (potential) drug targets. The selectivity compounds identified were checked by comparison with replication protein A. The assays appear to be robust. Appropriate controls are used. The discussion is informative.

Minor comments:

-A sentence in the abstract about the function of mcm10 would assist readers unfamiliar with this protein.

-In paragraph 2 of the introduction, the rationale for the choice of the mcm10 ID domain for screening is given, but it is not clear why the NTD was excluded.

-It would be useful for readers unfamiliar with the cell lines to state why HCT116 and hTERT-RPE1 cells were selected for cell assays.

-the abbreviation PPADS should be defined on first use in the text (p9). Did the authors mean to say "Interestingly, iso-PPADS and PPADS_failed to_inhibit probe binding to xMcm10-FL by > 50% up to a concentration of 2 mM"?

-From which species is the RPA used to assess inhibitor specificity?

-Supplementary figures should be cited consistently (compare "Supplementary Fig. S1" vs "Supplementary Fig. 5").

-The strange shapes of the inhibition curves in Figures S3b and S5 deserves comment. My guess is that the compounds are starting to precipitate at higher concentrations.

Review form: Reviewer 3**Recommendation**

Accept with minor revision (please list in comments)

Are each of the following suitable for general readers?

- a) **Title**
Yes
- b) **Summary**
Yes
- c) **Introduction**
Yes

Is the length of the paper justified?

Yes

Should the paper be seen by a specialist statistical reviewer?

No

Is it clear how to make all supporting data available?

Not Applicable

Is the supplementary material necessary; and if so is it adequate and clear?

Yes

Do you have any ethical concerns with this paper?

No

Comments to the Author

This is a concise manuscript that tells a very interesting story of an unrewarding drug screening campaign that will be a useful reference for other workers who try to find inhibitors of protein-DNA interactions. The work is well done, the approach is well described and the results present an instructional narrative of interest to a broad readership. The authors describe the setup of a fluorescence polarization high-throughput screening (FP-HTS) assay for inhibitors of binding of the key DNA replication protein MCM10 to DNA. Human MCM10 is established as a potential anticancer target.

The FP-HTS and immediate follow up assays identified only the polysulfated polyaromatic urea compound suramin as being worthwhile of being pursued. The authors then followed up with high-quality affinity (FP and SPR) and cell-based assays to examine a range of suramin analogs. The conclusion is that suramin and its relatives act as DNA analogs to compete with the DNA-binding site in MCM10.

Suramin has turned up again and again as an inhibitor of many enzymes over the past 80 years or so, many of which are not known to interact with DNA (e.g. jack bean urease!). So although it is used clinically, its mechanisms and targets are not rigorously known. The authors could briefly review some of this fascinating literature to increase the general interest of their study.

There are a few typos and confusing sentences in the manuscript and more in the SI. I encourage the authors to read the m/s and SI carefully and make appropriate corrections before submission of a revised version..

Decision letter (RSOB-19-0117.R0)

04-Jul-2019

Dear Dr Bielinsky

We are pleased to inform you that your manuscript RSOB-19-0117 entitled "The anti-parasitic agent suramin and several of its analogs are inhibitors of the DNA binding protein Mcm10" has been accepted by the Editor for publication in Open Biology. The reviewer(s) have recommended publication, but also suggest some minor revisions to your manuscript. Therefore, we invite you to respond to the reviewer(s)' comments and revise your manuscript.

Please submit the revised version of your manuscript within 7 days. If you do not think you will be able to meet this date please let us know immediately and we can extend this deadline for you.

- 1) A text file of the manuscript (doc, txt, rtf or tex), including the references, tables (including captions) and figure captions. Please remove any tracked changes from the text before submission. PDF files are not an accepted format for the "Main Document".
- 2) A separate electronic file of each figure (tiff, EPS or print-quality PDF preferred). The format should be produced directly from original creation package, or original software format. Please note that PowerPoint files are not accepted.
- 3) Electronic supplementary material: this should be contained in a separate file from the main text and meet our ESM criteria (see <http://royalsocietypublishing.org/instructions-authors#question5>). All supplementary materials accompanying an accepted article will be treated as in their final form. They will be published alongside the paper on the journal website and posted on the online figshare repository. Files on figshare will be made available approximately one week before the accompanying article so that the supplementary material can be attributed a unique DOI.

Online supplementary material will also carry the title and description provided during submission, so please ensure these are accurate and informative. Note that the Royal Society will not edit or typeset supplementary material and it will be hosted as provided. Please ensure that the supplementary material includes the paper details (authors, title, journal name, article DOI). Your article DOI will be 10.1098/rsob.2016[last 4 digits of e.g. 10.1098/rsob.20160049].

- 4) A media summary: a short non-technical summary (up to 100 words) of the key findings/importance of your manuscript. Please try to write in simple English, avoid jargon, explain the importance of the topic, outline the main implications and describe why this topic is newsworthy.

Images

Data-Sharing

It is a condition of publication that data supporting your paper are made available. Data should be made available either in the electronic supplementary material or through an appropriate repository. Details of how to access data should be included in your paper. Please see <https://royalsocietypublishing.org/rsob/for-authors#question3> for more details.

Data accessibility section

Sincerely,

The Open Biology Team

<mailto:openbiology@royalsociety.org>

Reviewer(s)' Comments to Author:

Referee: 1

Comments to the Author(s)

Suramin is an interesting compound that acts as a DNA or RNA mimic and it is turning out that suramin can be used to block several DNA protein interactions. What I like about this paper is that they do careful characterizations of the suramin/analog interactions with the target. The anticancer activities are a bit weak but that could be because of permeation issues. In any case I recommend acceptance.

Referee: 2

Comments to the Author(s)

This is an interesting paper. DNA-binding proteins are considered "difficult" targets for drug discovery, so the success reported here in the identification of hits against mcm10 (albeit with a low hit rate) will be of wide interest to readers interested in such (potential) drug targets. The selectivity compounds identified were checked by comparison with replication protein A. The assays appear to be robust. Appropriate controls are used. The discussion is informative.

Minor comments:

-A sentence in the abstract about the function of mcm10 would assist readers unfamiliar with this protein.

-In paragraph 2 of the introduction, the rationale for the choice of the mcm10 ID domain for screening is given, but it is not clear why the NTD was excluded.

-It would be useful for readers unfamiliar with the cell lines to state why HCT116 and hTERT-RPE1 cells were selected for cell assays.

-the abbreviation PPADS should be defined on first use in the text (p9). Did the authors mean to say "Interestingly, iso-PPADS and PPADS failed to inhibit probe binding to xMcm10-FL by > 50% up to a concentration of 2 mM"?

-From which species is the RPA used to assess inhibitor specificity?

-Supplementary figures should be cited consistently (compare "Supplementary Fig. S1" vs "Supplementary Fig. 5").

-The strange shapes of the inhibition curves in Figures S3b and S5 deserves comment. My guess is that the compounds are starting to precipitate at higher concentrations.

Referee: 3

Comments to the Author(s)

This is a concise manuscript that tells a very interesting story of an unrewarding drug screening campaign that will be a useful reference for other workers who try to find inhibitors of protein-DNA interactions. The work is well done, the approach is well described and the results present an instructional narrative of interest to a broad readership. The authors describe the setup of a fluorescence polarization high-throughput screening (FP-HTS) assay for inhibitors of binding of the key DNA replication protein MCM10 to DNA. Human MCM10 is established as a potential anticancer target.

The FP-HTS and immediate follow up assays identified only the polysulfated polyaromatic urea compound suramin as being worthwhile of being pursued. The authors then followed up with high-quality affinity (FP and SPR) and cell-based assays to examine a range of suramin analogs. The conclusion is that suramin and its relatives act as DNA analogs to compete with the DNA-binding site in MCM10.

Suramin has turned up again and again as an inhibitor of many enzymes over the past 80 years or so, many of which are not known to interact with DNA (e.g. jack bean urease!). So although it is used clinically, its mechanisms and targets are not rigorously known. The authors could briefly review some of this fascinating literature to increase the general interest of their study.

There are a few typos and confusing sentences in the manuscript and more in the SI. I encourage the authors to read the m/s and SI carefully and make appropriate corrections before submission of a revised version.

Author's Response to Decision Letter for (RSOB-19-0117.R0)

See Appendix A.

Decision letter (RSOB-19-0117.R1)

22-Jul-2019

Dear Dr Bielinsky

We are pleased to inform you that your manuscript entitled "The anti-parasitic agent suramin and several of its analogs are inhibitors of the DNA binding protein Mcm10" has been accepted by the Editor for publication in Open Biology.

Article processing charge

Please note that the article processing charge is immediately payable. A separate email will be sent out shortly to confirm the charge due. The preferred payment method is by credit card; however, other payment options are available.

Sincerely,

The Open Biology Team
mailto: openbiology@royalsociety.org

Appendix A

David Glover, FRS
Editor-in-Chief
Open Biology

July 17, 2019

Dear Dr. Glover,

Thank you for the positive decision letter from July 4, 2019. We have enclosed a revised version of our manuscript entitled “The anti-parasitic agent suramin and several of its analogs are inhibitors of the DNA binding protein Mcm10”. I am submitting the paper on behalf of all authors: Carolyn N. Paulson, Kristen John, Ryan M. Baxley, Fredy Kurniawan, Kayo Orellana, Rawle Francis, Alexandra Sobeck, Brandt F. Eichman, Walter J. Chazin, Hideki Aihara, Gunda I. Georg, Jon E. Hawkinson, and Anja-Katrin Bielinsky. Dr. Hawkinson and I are co-corresponding authors on this manuscript.

We have amended our manuscript in response to the reviewers’ suggestions. Here, I briefly summarize the changes we have made:

- 1) We have added a sentence in the abstract to better describe that Mcm10 is essential for DNA unwinding by the replisome.
- 2) We have added a sentence to the 2nd paragraph (page 4) of the introduction to point out that the N-terminal domain of Mcm10 does not have any DNA binding activity.
- 3) We have included a rationale on page 11 for selecting the specific cell lines we utilized to test cytotoxic activity. Both are of epithelial origin.
- 4) We included the full name of PPADS, and revised a sentence on page 9 that was unclear. It now reads: “Interestingly, *iso*-pyridoxalphosphate-6-azophenyl-2', 4'-disulfonic acid (*iso*-PPADS) and PPADS inhibited probe binding to xMcm10-FL by less than 50% when added up to a concentration of 2 mM (Table 1 and Supplementary Fig. S3B).”
- 5) We make clear that human RPA was utilized.
- 6) We consistently cite Supplementary Figures as “Supplementary Figure S” followed by a number.
- 7) We have added to the figure legends for Supplementary Figures S3 and S5 and explanation for the “strange” shape of the inhibition curves. It reads: “The tailing up of the dose-response curves for both compounds at concentrations of 1 mM could be due to lack of solubility, a detergent-like effect of these amphipathic molecules, or fluorescence interference.”
- 8) We have also added a little bit more information on suramin as suggested by reviewer 3. We included recent findings of the beneficial effects of low-dose suramin in the treatment of autism-like disorders (references 37-39 on page 12).
- 9) We have corrected typos and revised wording for clarity in both the main manuscript and the supplementary information.

On behalf of all authors, I'd like to thank you very much for serving as our editor, and soliciting very constructive reviews. We believe that the review process has further strengthened our manuscript.

Sincerely,

Anja-Katrin Bielinsky, Ph.D.

Professor

Department of Biochemistry, Molecular Biology & Biophysics

University of Minnesota Medical School

6-155 Jackson Hall

321 Church Street SE

Minneapolis, MN 55455

612-624-2469 Office Phone

612-625-2163 Fax

bieli003@umn.edu